# *Histoplasma capsulatum* in wild mammals from Ecuador

Fernanda Hernández-Alomía[1☉], Jorge Brito[1☉], Ana Lucia Pilatasig[2], Daniela Reyes-Barriga[1], Julio C. Carrión-Olmedo[1], Pablo Jarrín-V[1], Pablo Sánchez[1], Santiago F. Burneo[2], Maria Alejandra Camacho[2], David Vasco-Julio[3], Manuel Calvopiña[4], Jacobus H. De Waard[4], Daniel Romero-Alvarez[1,5☉], Carlos Bastidas-Caldes[ID][6☉]*

1 Instituto Nacional de Biodiversidad (INABIO), Quito, Ecuador, 2 Museo de Zoología, Colección de Mastozoología, Pontificia Universidad Católica del Ecuador, Quito, Ecuador, 3 Laboratorio de Biotecnología Veterinaria, Autopista Manuel Córdoba Galarza, Calacalí, Ecuador, 4 One Health Research Group, Universidad de las Américas, Quito, Ecuador, 5 Research Group of Emerging and Neglected Diseases, Ecoepidemiology and Biodiversity, Health Science Faculty, School of Biomedical Sciences, Universidad Internacional SEK (UISEK), Quito, Ecuador, 6 School of Medicine, Universidad Espíritu Santo, Samborondón, Ecuador

☉ These authors contributed equally to this work.
* cabastidasc@gmail.com

## Abstract

### Background

*Histoplasma capsulatum* is a globally distributed dimorphic fungal pathogen endemic to the Americas, Africa, and Asia. It causes histoplasmosis, a disease acquired via inhalation of spores from contaminated environments. It thrives in nitrogen-rich soils and is disseminated by avian and chiropteran reservoirs. *Histoplasma capsulatum* has been found in wild mammals such as rodents, marsupials, felines, and xenarthrans, suggesting diverse reservoirs that may influence its maintenance and transmission in endemic areas. This study aimed to detect *H. capsulatum* in tissues of wild small mammals sampled across Ecuador.

### Methods

Tissue samples (n = 324) were collected from wild mammals across the Coast, Andean, and Amazon regions between 2022 and 2023. Species were identified morphologically and *H. capsulatum* was detected using nested PCR targeting the 100-kDa protein-encoding gene. Positive samples were sequenced and analyzed. Ecological niche modeling focused on environmental clustering, via one class support vector machine (OC-SVM) hypervolumes, identified regions suitable for fungal survival.

provided the original author and source are credited.

**Data availability statement:** All relevant data are in the manuscript and its supporting information files.

**Funding:** This study was funded by the Universidad de Las Américas (UDLA) under Research Project 486.B.XIV.24, directed by CBC. (Funder website: https://www.udla.edu. ec). The funder had no role in the study design, data collection and analysis, decision to publish, or preparation of the manuscript.

**Competing interests:** The authors have declared that no competing interests exist.

## Results

*H. capsulatum* was detected in 14% of samples in 30 of 106 species studied, predominantly in Chiroptera (80%) followed by Rodentia (15%) and Didelphimorphia (4%). Suitable environmental conditions were concentrated in Ecuador's Coast region with isolated patches in the Andean and Amazon regions.

## Conclusion

This study documents the broad host range and ecological distribution of *H. capsulatum* in Ecuador, reinforcing concerns about its zoonotic potential. The detection of the fungus across diverse mammalian taxa and ecosystems emphasizes the importance of wildlife-based surveillance to better understand fungal pathogen reservoirs and geographic hotspots.

### Author summary

Histoplasmosis is a fungal disease that humans can get by breathing tiny spores from soil or animal droppings. The fungus, *Histoplasma capsulatum*, grows in nutrient-rich environments and is often linked to bats and birds. However, little is known about how it persists in wildlife in Ecuador. In this study, we collected tissue samples from small wild mammals across Ecuador's Coast, Andes, and Amazon regions. Using DNA-based methods, we detected *Histoplasma* in several species, with most positive animals being bats, followed by rodents and opossums. We also analyzed environmental conditions and identified regions of Ecuador where the fungus is most likely to survive. Our findings show that *Histoplasma* is widespread in different mammal species and ecosystems, which highlights the importance of wildlife monitoring to better understand where this fungus lives and to reduce the risk of disease in people living in endemic areas.

## Introduction

*Histoplasma* is a genus representing a worldwide thermal dimorphic fungal pathogen mainly found across the Americas, Africa, and Asia [1]. One of the most important species of this genus, *H. capsulatum*, is considered a causative agent of histoplasmosis, which is acquired by inhalation of spores from contaminated soil. The infectious cycle involves the inhalation of infective mycelial spores, which undergo a thermally regulated transition to the yeast form upon entering the host. This dimorphic shift is essential for intracellular survival and replication within macrophages and other phagocytes and is followed by the dissemination of the fungus back to the environment through carcasses, urine, and feces of infected animals [2]. It is suspected that histoplasmosis outbreaks may be attributable to environmental disruption, climate change, and human land use [3]. *Histoplasma capsulatum* is known to grow

in nitrogen and phosphate enriched environments, such as those with bird and bat guano [4]. Moreover, it can be dispersed by avian and chiropteran species, which might act as long-range migratory reservoirs [4].

South and Central America contributes to the majority of human and animal histoplasmosis cases worldwide. Brazil has made significant contributions to histoplasmosis research, including reports of *H. capsulatum* isolation from various animals and soil samples across seven states [5–7]. Histoplasmosis is prevalent in wild animals, including rodents, marsupials, felines, xenarthrans, and others, as well as domestic animals such as bovines, equines, sheep, dogs, and cats [8,9]. Apart from Brazil, little research has been done in Latin America with limited reports on clinical or animal studies in countries like Colombia [10], Peru [11], Venezuela [12], and Mexico [13,14].

In Ecuador, histoplasmosis has been historically underdiagnosed, although an increasing number of clinical reports suggest a broader distribution than previously recognized. In people living with HIV, a retrospective study documented multiple cases of disseminated histoplasmosis, highlighting its importance as an opportunistic infection [15]. Several hospital-based studies have further described probable and histologically confirmed cases in immunocompetent individuals presenting with respiratory or systemic symptoms in various regions of the country [16–18]. A fatal pediatric histoplasmosis case was reported in 2023, involving prolonged fever, hemorrhagic diarrhea, and persistent anemia [19]. Moreover, a 2015 report described a dog presenting with multifocal lymphadenitis and mucocutaneous lesions, in which *H. capsulatum* was identified as the causative agent [20].

Notably, most human cases reported in South America are associated with rural settings with a heightened likelihood of contact with wildlife. Consequently, the World Health Organization (WHO) encourages countries to strengthen monitoring of wildlife diseases as approximately 70% have a zoonotic origin [21]. Despite the relevance of the animal interface for spillover events, no studies on wild animals have been performed for *H. capsulatum* in Ecuador.

The concept of Primary Pandemic Prevention (PPP), introduced in the aftermath of the COVID-19 pandemic, emphasizes the early detection and surveillance of pathogens in animal populations—particularly in wildlife—as a critical measure to prevent zoonotic spillovers [22]. This perspective is inherently aligned with the One Health approach, which advocates for the integrated management of human, animal, and environmental health to address the complex dynamics of emerging infectious diseases [23]. Within this framework, the zoonotic potential of *H. capsulatum* remains insufficiently characterized. The present study aims to contribute to this gap by identifying the presence of the fungus in tissues of wild mammals sampled across 19 provinces of Ecuador, spanning the Coast, Andean, and Amazon regions.

## Methodology

### Ethics statement and permits

All fieldwork and specimen collection were conducted in compliance with Ecuadorian environmental and bioethical regulations. The general collection was authorized under the framework contract for access to genetic resources, issued by the National Biodiversity Directorate (Dirección Nacional de Biodiversidad in spanish) of the Ministry of Environment, Water and Ecological Transition (MAATE) which also served as the Institutional Review Board overseeing the study. Bats were captured and euthanized following the protocols of the QCAZ Museum under permit MAAE-DBI-CM-2021–0165. Terrestrial rodents and other small non-volant mammals were collected and euthanized by INABIO, operating under permit MAAE-DBI-CM-2023–0334. In the case of armadillos, tissues were obtained from a previously authorized study (MAAE-DBI-CM-2021–0172) in which hunters donated armadillos' organs. In Ecuador, indigenous communities in rural regions can legally use bushmeat as protein sources for subsistence, a common practice that it is recognized under national regulations.

### Sample site and specimen collection

Ecuador is divided into three climatic regions: Coast, Andean, and Amazon regions. Sampling efforts were conducted throughout the country between January 2022 and August 2023. Sampling sites include protected natural areas, secondary forests—influenced by agricultural activity—and highly deforested areas in rural Ecuador.

Collection techniques varied depending on the order of mammals collected. For bats (Chiroptera), a total of 18 mist nets were installed for a period of ten consecutive nights in a variety of locations, including agricultural fields and secondary forests. The mist nets were placed at ground level in areas with a high probability of bat capture, such as foraging sites, roosts, streams, and caves. The mist nets were left open for seven hours each night.

The capture of non-volant mammals, comprising rodents (Rodentia), opossums (Didelphimorphia), rabbits (Lagomorpha), marsupial mice (Paucituberculata), and shrews (Eulipotyphla) was conducted in multiple locations within Ecuador. To capture animals, live traps—one hundred Sherman and 15 Tomahawk traps—were utilized along transects spanning approximately 200 meters. A pitfall trap system was utilized at each location to optimize the efficacy of trapping. This entailed the use of a line comprising ten buckets (20 L capacity) buried at ground level and crossed by a plastic barrier approximately 60 m long and 0.5 m high [24]. Sherman, Tomahawk, and pitfall traps were baited with a mixture of oats, vanilla extract, and coconut. Armadillo specimens (Cingulata) were procured directly from hunters in rural regions of Ecuador who donated armadillo organs as part of a different study [25].

## Morphological species identification and tissue sampling

The handling of captured individuals followed the protocols established by the Mammalogy Division of the Museum of Zoology of the Pontifical Catholic University of Ecuador (i.e., QCAZ) [26] and the guidelines of the American Society of Mammalogists [27,28]. Specimens were euthanized via cervical dislocation, adhering strictly to the ethical protocols outlined by Sikes et al. [28] and Erazo et al. [27]. Rodent and marsupial specimens were deposited at the Ecuadorian National Institute of Biodiversity (i.e., INABIO, in Spanish) in Quito, Ecuador. Identification of rodents was conducted following the methods described by Patton et al. [29], while marsupial identification adhered to the protocols of Voss and Giarla [30]. Armadillos were identified using molecular methods detailed in [25]. Collected specimens, along with tissue samples (liver, intestine, and muscle), were individually preserved in 75% ethanol to ensure proper conservation.

## Molecular procedures

The DNA extraction was conducted using glass milk methodology, a silica-based method which uses silicon dioxide ($SiO_2$) powder in a guanidine–HCl solution [31]. Ten milligrams of the target tissue were subjected to a preliminary treatment with Tris-HCl (10 mM, pH 8.0) and 20 mg/mL proteinase K, and then incubated overnight at 50°C. On the subsequent day, 200 µL of 2X concentrated glassmilk binding buffer and TE (10/1) were added to the sample and incubated for 20 minutes at 65°C. A volume of 200 µL of isopropanol was then added and the mixture was subjected to centrifugation at 12,000 rpm for one minute. The resulting pellet was resuspended in 500 µL of the Wash Buffer and centrifuged again at 12,000 rpm for 1 minute. This step was repeated twice. The pellet was then washed with 70% ethanol, centrifuged, and discarded. The remaining ethanol was allowed to dry at 65°C for 5–10 minutes until it completely evaporated. Finally, the pellet was resuspended in 200 µL of TE and stored overnight at 4°C. The supernatant was transferred to a new microtube and stored at -20°C for further use in molecular analyses [31].

For the identification of the 100-kDa protein-coding gene of *H. capsulatum*, a nested PCR was performed. For the first PCR, HcI (5-GCGTTCCGAGCCTTCCACCTCAAC-3) and HcII (5-ATGTCCCATCGGGCGCCGTGTAGT-3) primers were used to amplify a region of 391 base pairs (bp). For the second PCR, the primers used were HcIII (5-GAGATCTAGTCGCGGCCA GGTTCA-3) and HcIV (5-AGGAGAGAACTGTATCGGTGGCTTG-3), resulting in a PCR product of 210 bp. For this stage, the product of the first PCR was used as DNA template. Amplification was performed in reactions of 20 µL containing 1X GoTaq Green Master Mix (Promega) and 0.4 µM of each primer. The PCR protocol was followed as stated by Bialek et. al [32].

## Sequencing analysis and Identification

The second PCR products (210 bp) were sequenced using the Sanger technique on an ABI 3500xL Genetic Analyzer (Applied Biosystems, Foster City, CA, USA) with BigDye 3.1 Terminator and a capillary electrophoresis matrix. PCR products were purified by enzymatic purification with *Exo I* and FastAP [33]. Sequences were edited using MEGA X software

[34] and compared against sequences available at the GenBank database at the National Centre for Biotechnology Information (NCBI) through the Basic Local Alignment Search Tool (BLAST) [35]). Sequences were deposited in the NCBI database which assigned the following accession numbers: OR242319-OR242355; PP669556-PP669565.

## Statistical analysis

All statistical analyses were performed using R software (version 4.4.1). To determine whether the prevalence of *H. capsulatum* varied significantly among wild mammal taxonomic orders, a global Chi-square test of independence was applied. Further, we calculated Chi-square tests comparing one *H. capsulatum* positive order versus all the other positive orders combined. For both approaches, the null hypothesis assumed no association between host order and fungal detection status. Statistical significance was defined as $p < 0.05$.

## Potential distribution of *H. capsulatum*

We used ecological niche models to suggest potential regions of environmental similarity in the country using one-class support vector machine (OC-SVM) hypervolumes as presence-only algorithm [36]. Occurrences of positive detections of *H. capsulatum* were combined with temperature, relative humidity, and chemical characteristics of soil as environmental predictors summarized with a principal component analysis (PCA) and using only those components collecting more than 85% of the variability (S1 Information). We modeled the potential environmental suitability of the pathogen assuming that positive detections in wildlife act as sentinels of environmental fungi spores; thus, the model represents regions of environmental suitability for *H. capsulatum* spore survival in Ecuador.

## Results

A total of 325 tissue samples—including 309 liver, seven muscle, three kidney, three fecal, and two spleen samples, plus a unique lung sample—were collected from wild mammals belonging to at least seven taxonomic orders and 106 species across the Ecuadorian Coast, Andes, and Amazon regions between 2022–2023 (S1 Table). The samples collected included 182 individuals from the order Rodentia, 70 from Chiroptera, 37 from Cingulata, 19 from Paucituberculata, 14 from Didelphimorphia, two from Eulipotyphla, and one from Lagomorpha.

The geographic distribution of mammalian specimens encompassed the three ecological regions of Ecuador (Coast, Andean, and Amazon) providing comprehensive nationwide coverage. The highest sampling effort was concentrated in the Amazon and Andean regions, with Chiroptera representing the most collected specimens. Other mammalian orders, such as Didelphimorphia and Rodentia, were sampled in smaller numbers but distributed across the three regions as well (Fig 1).

Positive detections were identified as *H. capsulatum* through sequencing (S2 Table). Overall, *H. capsulatum* was present in 14.1% (n = 46/325) of the evaluated samples, distributed across the three regions of Ecuador. Positive cases were detected in three of the seven mammalian orders assessed, encompassing 28% (30/106) of the species examined and showing a statistical association between mammalian order and fungal detection ($x^2 = 86.056$, df = 2, $p < 0.0001$, S3 Table). Among these, Chiroptera demonstrated the highest prevalence, with 80% positive samples (n = 37/46; $x^2 = 62.839$, df = 2, $p < 0.0001$, S3 Table), followed by Rodentia with 15.22% positives (n = 7/46; $x^2 = 23.12$, df = 2, $p < 0.0001$, S3 Table) and Didelphimorphia with 4.34% positives (n = 2/46; $x^2 = 0.095$, df = 2, $p = 0.953$, S2 Table). Remarkably, a single positive case was detected from a wild mouse in the fluvial highlands of Cubilines, Chimborazo, in the Andean region. In contrast, no *H. capsulatum* was detected in Eulipotyphla, Lagomorpha, Cingulata, and Paucituberculata species.

Among the *H. capsulatum*-positive samples, the pathogen was detected in 20 out of 27 identified species of Chiroptera, seven out of 63 species of Rodentia, and two out of nine species of Didelphimorphia (Tables 1 and S1). Additionally, two positive bat specimens could not be taxonomically identified using available classification keys. Notably, this study provides the first documented evidence of *H. capsulatum* infection in two opossum species, six rodent species, and 14 bat

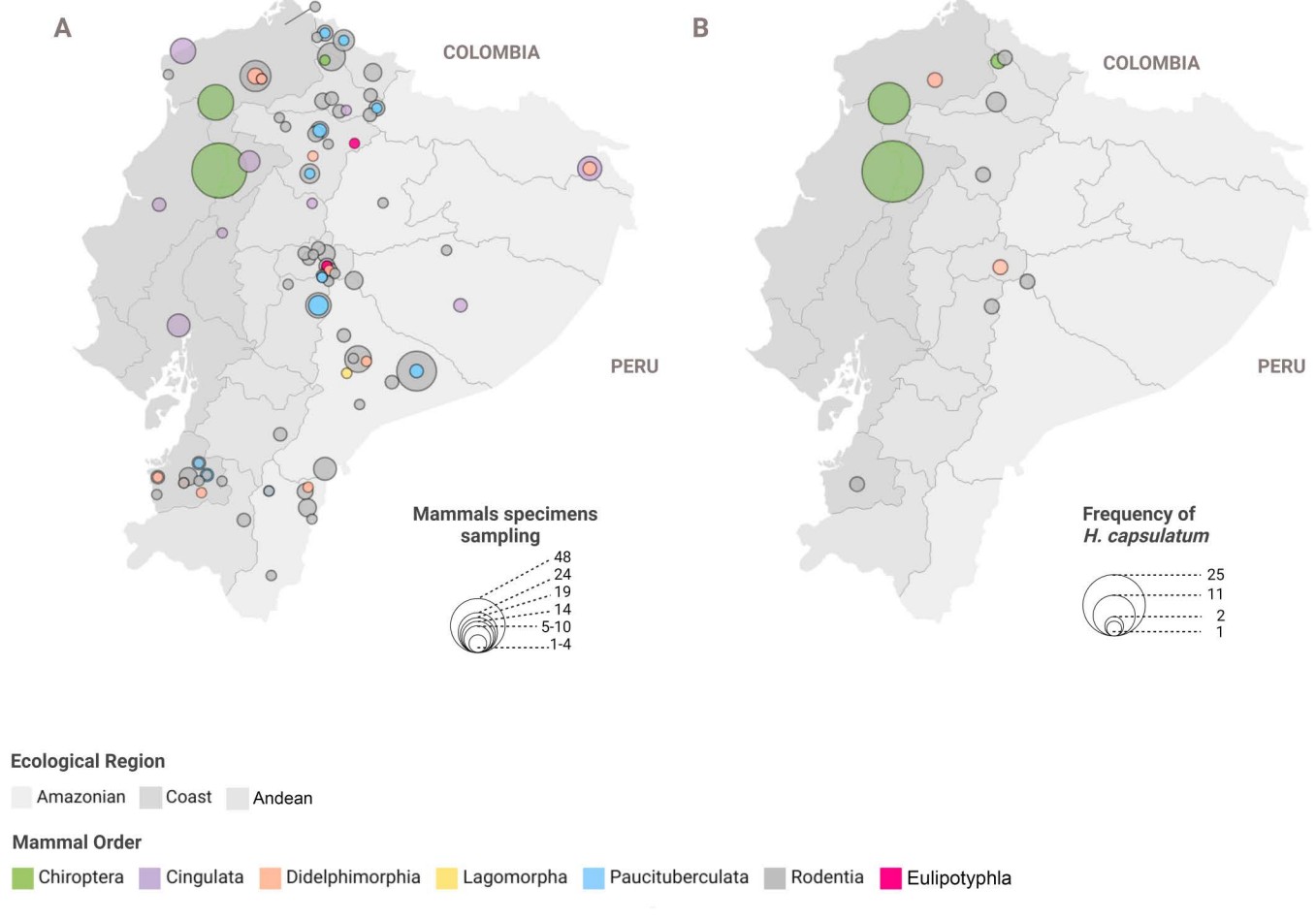

**Fig 1. Distribution of examined mammalian species in Ecuador across ecological regions.** Panel A represents the sampling magnitude of mammal specimens across the Amazon rainforest, Andean highlands, and Coast lowlands. Panel B illustrates the frequency of *Histoplasma capsulatum* detected in mammalian tissues within these regions. Ecuador, located on the equatorial line northwest of South America, provides this study's unique geographic and ecological context.

species in Ecuador, incriminating newer wild species and expanding the known host range of this pathogen. A comprehensive summary of the species, geographic regions, altitudes, and tissue types associated with positive *H. capsulatum* detections is detailed in Table 1.

The detection of *H. capsulatum* exhibited marked spatial variation, with the highest frequencies—up to 25 positive cases per site—observed in the northern Coast and specific areas of the Andean region. Lower detection frequencies, ranging from 1 to 11 cases, were recorded in the Amazon. Details regarding the sampling effort magnitude and the frequency of positive detections for *H. capsulatum* are provided in Fig 1 and S1 Table.

Ecuadorian regions with environments similar to those where *H. capsulatum*-positive cases were detected are primarily concentrated in the Coast region, northern Ecuador, where the largest area of continuous suitability was predicted (Fig 2). Additional areas of continuous suitability extend into parts of the Andean region, particularly in its western and central zones, as well as in the Coast lowlands. Discontinuous, patchy areas of suitability were also identified in the southern Coast region and in the eastern portion of the Amazon, where fragmented areas of predicted presence were observed (Fig 2).

**Table 1. Prevalence of *Histoplasma capsulatum* in wild mammal hosts from Ecuador. Acronyms: masl=meters above sea level.**

| Order (%; n/N) | Positive host species (n) | Prevalence (%) | Source/ tissue | Elevation (masl) | Ecuadorian region | Previous reports Country (Reference) |
|---|---|---|---|---|---|---|
| Didelphimorphia 14.3 (2/14) | *Marmosa germana** (1) | 100 (1) | Liver | 2750 | Andean | N/A |
| | *Monodelphis sp.** (7) | 1/7 | Liver | 450 | Coast | N/A |
| Rodentia 3.85 (7/182) | *Handleyomys alfaroi** (3) | 33.3 (1) | Liver | 449 | Coast | N/A |
| | *Hylaeamys tatei** (2) | 50(1) | Liver | 1284 | Amazonia | N/A |
| | *Microryzomys minutus** (11) | 9.1(1) | Liver | 2919 | Andean | N/A |
| | *Neomicroxus latebricola** (2) | 50(1) | Liver | 2956 | Andean | N/A |
| | *Oecomys* sp. (7) | 14.3(1) | Liver | 1262 | Andean | Brazil [5] |
| | *Oligoryzomys sp.** (1) | 100 (1) | Liver | 2956 | Andean | N/A |
| | *Thomasomys paramorum** (12) | 8.3 (1) | Liver | 3893 | Andean | N/A |
| Chiroptera 53.62 (37/69) | *Artibeus fraterculus** (1) | 100 (1) | Liver | 193 | Coast | N/A |
| | *Artibeus literatus* (2) | 50(1) | Liver | 200 | Coast | Brazil [6,7] |
| | *Artibeus ravus** (4) | 50(2) | Liver, feces | 200 | Coast | N/A |
| | *Carollia brevicaudum** (12) | 50(6) | Liver | 193 - 200 | Coast | N/A |
| | *Carollia castanea** (6) | 50(3) | Liver | 193 - 200 | Coast | N/A |
| | *Carollia perspicillata* (6) | 83.3 (5) | Liver | 200 | Coast | Colombia, Panama, Brazil, Trinidad [37–43] |
| | *Eptesicus innoxius** (1) | 100 (1) | Liver | 200 | Coast | N/A |
| | *Glossophaga soricina* (6) | 50 (3) | Liver | 193 | Coast | Panama, Brazil, Trinidad [6,38–40,42,43] |
| | *Lonchophylla concava** (1) | 100 (1) | Liver | 200 | Coast | N/A |
| | *Lonchorhina aurita* (1) | 100 (1) | Liver | 200 | Coast | Panama [39,42] |
| | *Micronycteris* cf. Giovanniae* (1) | 100 (1) | Liver | 1100 | Andean | N/A |
| | *Myotis nigricans* (4) | 25 (1) | Liver | 193 | Coast | Brazil [6] |
| | *Phyllostomus discolor* (1) | 100 (1) | Liver | 200 | Coast | Panama, El Salvador [39,43] |
| | *Platyrrhinus dorsalis* cf.* (1) | 100 (1) | Liver | 200 | Coast | N/A |
| | *Platyrrhinus umbratus** (4) | 25 (1) | Liver | 200 | Coast | N/A |
| | *Rhinophylla alethina** (3) | 66.6 (2) | Liver | 200 | Coast | N/A |
| | *Sturnira bakeri* cf.* (1) | 100 (1) | Liver | 193 | Coast | N/A |
| | *Sturnira luisi** (1) | 100 (1) | Liver | 200 | Coast | N/A |
| | *Uroderma convexum** (1) | 100 (1) | Liver | 200 | Coast | N/A |
| | *Vampyressa thyone** (1) | 100 (1) | Liver | 200 | Coast | N/A |
| | Unidentified (3) | 66.6 (2) | Liver, feces | 193-200 | Coast | N/A |

Note: * Indicates species in which Histoplasma capsulatum is reported for the first time.

## Discussion

This study presents compelling evidence for the occurrence of *H. capsulatum* in wild mammals distributed across the three Ecuadorian continental regions. The analysis revealed a prevalence of 14% among the wild mammal samples examined. In this study, *H. capsulatum* was identified in 30 of the106 individual species studied.

The identification of bats, rodents, and marsupials as carriers of this pathogen is paramount. The elevated prevalence of *H. capsulatum* among Chiroptera (80%) highlights their potential role as primary reservoirs, particularly in anthropogenically disturbed forested habitats. We found a statistically significant association between Chiroptera and *H. capsulatum* detection ($\chi^2=62.84$, $p<0.0001$), indicating that bats are more affected compared to other mammalian groups.

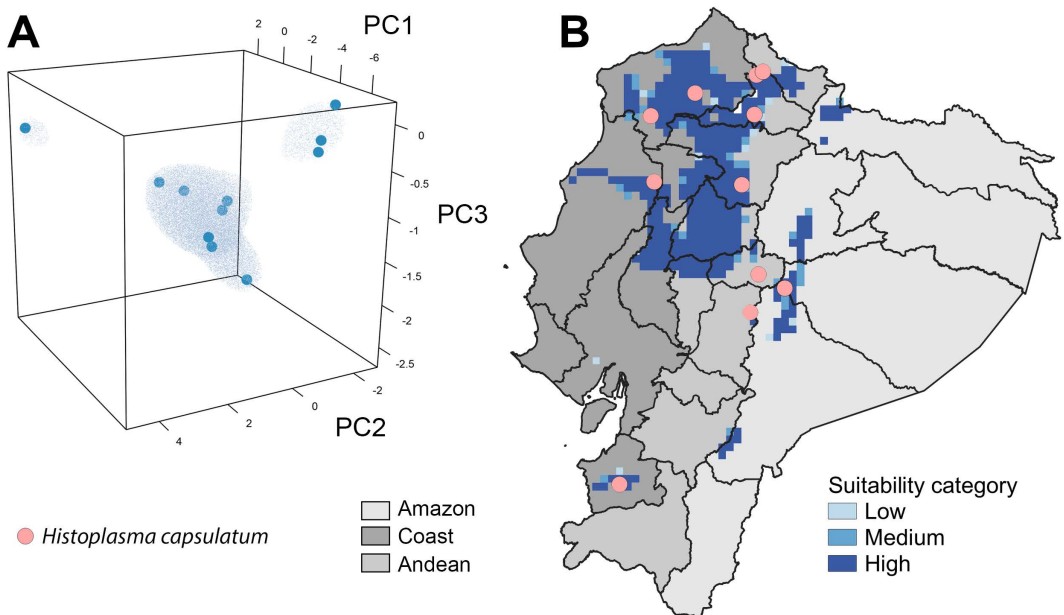

**Fig 2. Potential distribution of *Histoplasma capsulatum* based on environmental clustering.** Environmental suitability was modeled using one-class support vector machine (OC-SVM) hypervolumes to identify clusters in environmental space (A), represented by principal components (PC1, PC2, and PC3), and later projected onto the geographic landscape of Ecuador (B). The geographic projection highlights suitability categories (low, medium, and high) across regions, with high suitability areas primarily concentrated in northern Ecuador, spanning the Coast and Andean regions. Pink dots indicate locations where *H. capsulatum* was detected. Figure was developed using the packages alphahull and ggplot2 in R programming language and the official shapefile of Ecuador developed by the Technical Secretary of the National Committee on Internal Limits (CONALI in Spanish), available at: https://www.gob.ec/conali".

Of the 27 bat species sampled, 20 were identified as reservoirs of *H. capsulatum* (Table 1). Previous studies have documented multiple Chiroptera species as reservoirs [5,44]. Notably, this study reports the detection of *H. capsulatum* in 13 additional bat species, predominantly concentrated in the Coast and Andean regions of Ecuador. The high diversity of bat species harboring *H. capsulatum* suggests the potential existence of additional, yet unidentified, wildlife reservoirs for this pathogen.

In the Americas, *H. capsulatum* has been described previously in three of the host orders studied here, including Chiroptera [14,37], Rodentia [45,46], and Didelphimorphia [5,47]. In previous studies, *H. capsulatum* has been found in the viscera of *Didelphis albiventris*, a widely distributed marsupial, and in liver and spleen samples from introduced rats (*Rattus rattus*) and two other marsupials (*Metachirus opossum*) in Rio de Janeiro [5]. *H. capsulatum* was also isolated from captive maras (*Dolichotis patagonum*) in Mexico, with soil samples from their cages identified as the probable source of infection. In our study, despite Didelphimorphia was found positive for *H. capsulatum*, the association with fungal status was not statistically significant ($\chi^2 = 0.096$, $p = 0.953$) in comparison with the other orders assessed, suggesting that it might be less important as reservoir for *H. capsulatum*. On the other hand, Rodentia showed a significant association with fungal detection ($\chi^2 = 23.12$, $p < 0.0001$), suggesting a higher likelihood of infection in this group, possibly due to their wide distribution, high adaptability, and the social behavior observed in the majority of wild mice species, characteristics shared by many bats [48]. We detected *H. capulatum* in the wild mouse *Thomasomys paramorum* at 3,893 masl (Table 1 and Fig 2), which is a theoretical atypical environment for spore survival. However, *H. capsulatum* has been identified in colder environments, specifically it has been detected in soil and penguin excreta in the Antarctic Peninsula [49], Both arguments hint over the potential remarkable adaptability of *H. capsulatum* for environments and reservoirs.

Although *H. capsulatum* has been previously reported in armadillo tissues in studies conducted in Brazil [50,51], the samples in our analysis were negative for the pathogen. Furthermore, a related fungus, *Paecilomyces lilacinus*, was isolated from the nine-banded armadillo (*Dasypus novemcinctus*) and initially resembled *H. capsulatum* in impression films and smears in a study of 1984 [52]. These findings highlight the importance of considering various animal species as potential reservoirs for *H. capsulatum* and related fungi in different regions [5,43,48].

Most of *H. capsulatum* detections were identified in intervened forest areas near paddocks which might have increased the risk of spore dispersal to humans and wildlife. Despite the paucity of detailed information regarding human infections with *H. capsulatum* from bats in Ecuador, some cases have documented instances of infection among individuals who have visited caves with large bat colonies in the Amazon region [53]. Furthermore, a fatal case of a children infected in the coastal tropical region of Ecuador was published and initially misdiagnosed as visceral leishmaniasis. The immunocompromised patient had a history of exposure to guano in the vicinity of fruit trees, underscoring the potential for environmental reservoirs to contribute to misdiagnose fungal infections, particularly in vulnerable populations [20].

The algorithm used in the ecological niche model developed here can find environmental clusters [36] which is a step forward in classic approaches to spatial epidemiology which only focuses on geographic distances [54]. We leverage *H. capsulatum* identifications to suggest future regions of sampling considering the lack of studies of *H. capsulatum* in Ecuador and the limited resources to study fungi or other pathogens in wildlife. The model suggests that Ecuador's northern Andean and Coast region might be a pivotal area of environmental suitability for *H. capsulatum.* Identifications of the pathogen in the provinces of central Andes were neglected from the overall model suitability surface despite the development of 11 different predictions (S1 Information). Due to the scarcity of data and the inherent difficulties of detecting pathogens in wildlife, our models can be used as a first attempt to lead evidence-based sampling [25]. Species distribution models will greatly benefit from future field studies testing in-silico predictions such as the one presented here.

## One health perspectives

The identification of *H. capsulatum* across multiple mammalian orders highlights its zoonotic potential and underscores the importance of monitoring wildlife as sentinels for environmental pathogens. Thus, Primary Pandemic Prevention (PPP) is particularly relevant in this context, emphasizing the importance of proactive surveillance of fungal pathogens. Given that histoplasmosis often mimics other pulmonary and systemic diseases, misdiagnosis can result in significant morbidity and mortality, particularly in immuno-compromised populations [9].

Bats, as key reservoirs of *H. capsulatum*, play an integral role in trophic chains and provide essential ecosystem services, such as pollination, seed dispersal, and insect population control, which are vital for the health and sustainability of forests. However, areas where bat guano accumulates, such as caves or community dwellings near roots, pose a significant risk for the transmission of *H. capsulatum* in humans. To minimize these risks, public health strategies should include educational programs emphasizing safe practices when visiting caves for tourism or other activities. These practices could involve the use of protective gear, proper ventilation in guano-rich environments, and limiting access to sensitive areas to reduce spore dispersal.

## Study limitations

This study has several limitations. First, species sampling across all taxa is limited given constraints related to the conservation status of certain species, their vulnerability, restricted access to remote areas, and the regulatory requirements for collection permits. Accordingly, collection protocol prevented the capture or euthanasia of pregnant females or juvenile mammals. Despite these limitations, our findings provide valuable insights into the prevalence of *H. capsulatum* across a broad diversity of wild mammalian hosts (i.e., 106 species).

A further limitation was the inability to perform lineage-level identification of *H. capsulatum* via multilocus sequence typing (MLST). Because the tissue samples originated from preserved museum specimens and opportunistic field

collections, fungal culture was unfeasible, thereby excluding the use of housekeeping genes required for MLST analyses. Future studies should prioritize the use of fresh samples and fungal isolation to enable genomic characterization and a deeper understanding of the evolutionary dynamics and transmission ecology of *H. capsulatum* in wildlife. Finally, the ecological niche model presented here should be considered as preliminary considering the lack of systematic sampling of any of the mammalian orders. However, by using this methodology we are taking advantage of bioinformatic analyses to predict regions of future collection studies specifically aimed at recovering *H. capsulatum* spores from the environment.

## Conclusion

This study documents the broad host range and distribution of *H. capsulatum* in Ecuador, reinforcing concerns about its zoonotic potential. The detection of the fungus across diverse mammalian taxa and ecosystems emphasizes the importance of wildlife-based surveillance to better understand fungal pathogen reservoirs and geographic hotspots.

## Supporting information

**S1 Information.  Detailed methods of ecological niche modeling.**
(DOCX)

**S1 Table.  Complete dataset of host species and fungal detection results.**
(XLSX)

**S2 Table.  Accession numbers and sequences of *H. capsulatum* positive samples.**
(XLSX)

**S3 Table.  Chi-square Test.** Contingency tables and expected and observed frequencies by host orders.
(XLSX)

## Author contributions

**Conceptualization:** Pablo Sánchez, Jacobus H. de Waard, Carlos Bastidas-Caldes.

**Data curation:** Fernanda Hernández-Alomía, Daniel Romero-Alvarez.

**Formal analysis:** Fernanda Hernández-Alomía, Carlos Bastidas-Caldes.

**Funding acquisition:** Santiago F. Burneo, Carlos Bastidas-Caldes.

**Investigation:** Fernanda Hernández-Alomía, David Vasco-Julio, Carlos Bastidas-Caldes.

**Methodology:** Fernanda Hernández-Alomía, Jorge Brito, Ana Lucia Pilatasig, Daniela Reyes-Barriga, Julio C. Carrión-Olmedo, Pablo Jarrín-V, David Vasco-Julio, Carlos Bastidas-Caldes.

**Project administration:** Pablo Sánchez, Jacobus H. de Waard, Carlos Bastidas-Caldes.

**Resources:** Jorge Brito, Pablo Jarrín-V, Pablo Sánchez, Santiago F. Burneo, Maria Alejandra Camacho, Jacobus H. de Waard.

**Visualization:** Daniel Romero-Alvarez.

**Writing – original draft:** Manuel Calvopiña, Daniel Romero-Alvarez, Carlos Bastidas-Caldes.

**Writing – review & editing:** Santiago F. Burneo, Maria Alejandra Camacho, David Vasco-Julio, Manuel Calvopiña, Jacobus H. de Waard, Daniel Romero-Alvarez, Carlos Bastidas-Caldes.

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
