## [Decision Letter · Decision Letter 0]

12 May 2025

Histoplasma capsulatum in Wild Mammals from Ecuador

Dear Dr. Bastidas,

Thank you for submitting your manuscript to PLOS Neglected Tropical Diseases. After careful consideration, we feel that it has merit but does not fully meet PLOS Neglected Tropical Diseases's publication criteria as it currently stands. Therefore, we invite you to submit a revised version of the manuscript that addresses the points raised during the review process.

Please submit your revised manuscript within 60 days Jul 11 2025 11:59PM. If you will need more time than this to complete your revisions, please reply to this message or contact the journal office at plosntds@plos.org. Please include the following items when submitting your revised manuscript:

We look forward to receiving your revised manuscript.

Kind regards,

Angel Gonzalez, Ph.D.

Academic Editor

Marcio Rodrigues

Section Editor

Shaden Kamhawi

co-Editor-in-Chief

Paul Brindley

co-Editor-in-Chief

**Journal Requirements:**

At this stage, the following Authors/Authors require contributions: Carlos Bastidas. Please ensure that the full contributions of each author are acknowledged in the "Add/Edit/Remove Authors" section of our submission form.

- ® on page: 4 and 5.

5) We have noticed that you have uploaded Supporting Information files, but you have not included a list of legends. Please add a full list of legends for your Supporting Information files after the references list.

Potential Copyright Issues:

- Figure 1. Please confirm whether you drew the images / clip-art within the figure panels by hand. If you did not draw the images, please provide (a) a link to the source of the images or icons and their license / terms of use; or (b) written permission from the copyright holder to publish the images or icons under our CC BY 4.0 license. Alternatively, you may replace the images with open source alternatives. See these open source resources you may use to replace images / clip-art:

- Figures 2 and 3. Please (a) provide a direct link to the base layer of the map (i.e., the country or region border shape) and ensure this is also included in the figure legend; and (b) provide a link to the terms of use / license information for the base layer image or shapefile. We cannot publish proprietary or copyrighted maps (e.g. Google Maps, Mapquest) and the terms of use for your map base layer must be compatible with our CC BY 4.0 license.

**Reviewers' Comments:**

Reviewer's Responses to Questions

**Key Review Criteria Required for Acceptance?**

**Methods**

-Are the objectives of the study clearly articulated with a clear testable hypothesis stated?

-Is the study design appropriate to address the stated objectives?

-Is the population clearly described and appropriate for the hypothesis being tested?

-Is the sample size sufficient to ensure adequate power to address the hypothesis being tested?

-Were correct statistical analysis used to support conclusions?

-Are there concerns about ethical or regulatory requirements being met?

Reviewer #1: Phylogenetic analysis are not

Reviewer #2: Methods are not clear. Justification for collecting tissues (result.s site lung, liver, spleen and feces) but presenting on only liver in Table 1 is confusing. were other tissues not screened? Screened but not positive?

There is biased sampling , quantity of animals captured in one region may bias results.

Global Chi-square test may be inappropriate given the uneven distribution of species. what is the true abundance of the species in question? This test is assuming that you captured a representative set of host orders. wouldn't you test each order separately?

Approval for trapping and euthanasia was stated in manuscript

Reviewer #3: -Are the objectives of the study clearly articulated with a clear testable hypothesis stated? Yes

-Is the study design appropriate to address the stated objectives?

Comment: I would like to question the efficacy of the Hcp100 molecular marker for the phylogenetic analysis. The Hcp100 molecular marker is successfully used in clinical diagnosis and for searching the H. capsulatum presence in randomly infected animals, because this marker amplified a specific DNA fragment of 210 bp, in the second Nested-PCR reaction, which is unique for this fungal pathogen. Although its use for H. capsulatum phylogenetic analyses is not the most recommendable, mainly considering that the Hcp100 fragment presents very low polymorphism, clades obtained were certainly representative and discriminate the different groups of animals studied. Thus, I suggest including an appropriate interpretation in the Discussion section of the Manuscript.

-Is the population clearly described and appropriate for the hypothesis being tested? Yes

-Is the sample size sufficient to ensure adequate power to address the hypothesis being tested? Yes

-Were correct statistical analysis used to support conclusions? Please, see my comment

-Are there concerns about ethical or regulatory requirements being met? Yes

**Results**

-Does the analysis presented match the analysis plan?

-Are the results clearly and completely presented?

-Are the figures (Tables, Images) of sufficient quality for clarity?

Reviewer #1: Phylogenetic analysis are not

Reviewer #2: Percentage positivity is slightly misleading when only one representative of the species was captured.

Data in supplemental table doesn't seem to match Table 1 - what is the status of samples 294 and 317? is this collected from animal directly or was this from the cave floor? If the latter, this is an environmental sample

Due to the high similarity of these sequences and having only a single sequence from PCR, a phylogenetic tree is likely a poor methods to describe diversity. this is a ~180 bp fragment, so percent identity would be clearer. no branch support reported and provides little information on Histoplasma diversity in Ecuador.

The ecological niche model is not well described and could simply model species distribution rather that fungal distribution. Very confusing.

Reviewer #3: -Does the analysis presented match the analysis plan? Please, see my comment

-Are the results clearly and completely presented? Delete repetitive phrases in the Results section, I.e.: lines 240-242, - Sequences marked with an asterisk (PP669556, PP669558, OR242338) represent identical sequences found in other specimens analyzed in this study. Sequences marked with an asterisk (PP669556, 242 PP669558, OR242338) represent identical sequences found in other specimens analyzed in this study

-Are the figures (Tables, Images) of sufficient quality for clarity? Yes

**Conclusions**

-Are the conclusions supported by the data presented?

-Are the limitations of analysis clearly described?

-Do the authors discuss how these data can be helpful to advance our understanding of the topic under study?

-Is public health relevance addressed?

Reviewer #1: yes

Reviewer #2: The discussion is diffuse and sometimes confusing. The point that we know little about the range of potential infected animals is key, and I think the data collected support that. Focus on this. I think some statement of the limitations of the study and future directions would be useful.

Reviewer #3: -Are the conclusions supported by the data presented? Yes

-Are the limitations of analysis clearly described? Please, see my comment

-Do the authors discuss how these data can be helpful to advance our understanding of the topic under study? Please, see my comment

-Is public health relevance addressed? Yes

**Editorial and Data Presentation Modifications?**

Reviewer #1: (No Response)

Reviewer #2: There must be clearer description of methods, justification of tissues samples and improved statistical analysis, as suggested above.

Reviewer #3: Minor observation:

Delete repetitive phrases

I.e.: pages 240-242, - Sequences marked with an asterisk (PP669556, PP669558,

OR242338) represent identical sequences found in other specimens analyzed in

this study. Sequences marked with an asterisk (PP669556, 242 PP669558,

OR242338) represent identical sequences found in other specimens analyzed in

this study

**Summary and General Comments**

Reviewer #1: This manuscript addresses a highly relevant and timely topic by investigating Histoplasma capsulatum infection in a variety of wild mammalian hosts across Ecuador. The study provides valuable insights into the ecological distribution and potential zoonotic interfaces of H. capsulatum, contributing to ongoing discussions in medical mycology and One Health frameworks. However, while the core objectives are important, the manuscript still requires significant revisions to improve scientific rigor, methodological transparency, and interpretative depth. Several sections—including the abstract, introduction, methods, phylogenetic analysis, and discussion—would benefit from restructuring, clarification, and more robust contextualization within the existing literature. I outline specific comments and suggestions below to help the authors enhance the overall clarity, accuracy, and impact of their work.

Abstract:

Line 28 – The statement “is disseminated by avian and chiropteran reservoirs” appears to be overly speculative, particularly regarding avian hosts. While Histoplasma has been consistently detected in association with chiropteran reservoirs—supporting their role in its ecological dissemination—there is limited evidence implicating birds directly.

Line 29 – “H. capsulatum presence is prevalent in wild animals, including rodents, marsupials, felines, xenarthrans, and others.” The statement is vague and lacks a clear take-home message. While it lists various wild animal groups associated with H. capsulatum, it doesn't explain the ecological, epidemiological, or transmission relevance of this information. Suggestion: “Histoplasma capsulatum has been found in wild mammals such as rodents, marsupials, felines and xenarthrans, suggesting diverse reservoirs that may influence its maintenance and transmission in endemic areas. ”. Please connect this to the next sentence.

Line 34 – the term “protein gene” is not commonly used in formal scientific writing. “H. capsulatum was detected using a nested PCR assay targeting the 100-kDa protein-encoding gene”

Line 35 - Ecological niche modeling - Which Method?

Line 40 – Improve conclusion. Suggestion: “ This study documents the broad host range and ecological distribution of H. capsulatum in Ecuador, reinforcing concerns about its zoonotic potential. The detection of the fungus across diverse mammalian taxa and ecosystems emphasizes the importance of wildlife-based surveillance to better understand fungal pathogen reservoirs and geographic hotspots”

Introduction

49 – Given the existence of multiple phylogenetic species within the Histoplasma genus, it is important to avoid generalizing H. capsulatum as a unique species.

51 - The description of the infectious cycle omits a pivotal phase in the pathogenesis of Histoplasma spp.—the transition from the mycelial to the yeast form (dimorphism) upon entering the host. This thermally regulated shift is critical for intracellular survival and virulence, particularly within macrophages. I recommend revising the sentence to explicitly include this phase, as it represents a fundamental step in the fungal life cycle and host adaptation.

53 - the reintroduction of the fungus into the environment??

60 – The sentence referencing Brazil’s contributions to histoplasmosis research contains critical grammatical and structural issues. I recommend rephrasing it for clarity and accuracy.

68 – The manuscript would benefit from a more comprehensive overview of the burden of histoplasmosis in Ecuador. As currently written, the mention of a single case gives the misleading impression that histoplasmosis is rare or poorly relevant in the country. However, this contradicts existing evidence. For instance, the study 10.1007/s10096-017-2928-5 highlights multiple cases of disseminated histoplasmosis in Ecuadorian patients with HIV, underscoring its clinical importance.

Line 78 - The concept of Primary Pandemic Prevention (PPP), as described here, is important and timely. However, I suggest that framing this within a One Health context may be more appropriate and impactful. One Health emphasizes the interconnectedness of human, animal (both domestic and wild), and environmental health, and has been widely adopted as a guiding framework for pandemic preparedness and zoonotic disease surveillance.

Line 83 - The final paragraph of the introduction would benefit from a more impactful closing that clearly states the study’s main findings and the broader implications of the knowledge gap being addressed. I recommend adding a sentence that emphasizes both the novelty of detecting Histoplasma across a diverse range of wild mammalian hosts in Ecuador and the public health relevance of these findings.

Methodology

89 – 109

This section raises two important concerns that require clarification. Animal Handling and Necropsy Procedures: The manuscript does not provide sufficient detail on how animals were euthanized or how necropsies were performed. For studies involving the collection of tissue samples from wild mammals, it is essential to specify the methods used for euthanasia (if applicable), as well as the procedures and biosafety measures applied during necropsy. This is critical for reproducibility, ethical transparency, and to ensure compliance with international animal welfare standards. Legality of Armadillo Acquisition: The collection of armadillo specimens from hunters or rural markets raises potential ethical and legal concerns. In many countries, including Ecuador, hunting and trade of wild mammals—especially for bushmeat—may be prohibited or tightly regulated. The authors should clarify whether this practice was conducted legally and under appropriate authorization. If specimens were sourced without proper documentation, this could present a serious ethical issue.

131 - a silica-based technique for nucleic acid purification?

134 - What is Magic Blue???

134-136 – Could the authors clarify whether DNA precipitation and wash was carried out at room temperature or at a lower temperature (e.g., 4 °C or −20 °C), which is standard to enhance nucleic acid yield?

137 - Could the authors specify the composition of the wash buffer used in the DNA extraction protocol?

148 - Please rephrase the sentence describing the PCR setup for clarity.

154 - The sentence refers to “BigDye 3.1® capillary electrophoresis matrix,” but this is technically inaccurate. BigDye Terminator v3.1® refers to the sequencing chemistry, not the electrophoresis matrix. Please revise the wording to accurately reflect that BigDye is the reagent used for Sanger sequencing, while the capillary electrophoresis matrix (e.g., POP-7 or POP-6) refers to the polymer used during fragment separation. Moreover, how PCR products were purified?? What is the size of the sequenced amplicons???? A lot of information is missing here.

Line 157 – It is unclear which sequences were used for comparison in the phylogenetic or identity analyses. I strongly recommend including a supplementary table listing all reference sequences used, along with their GenBank accession numbers, species identification, and relevant metadata.

Line 177 - The manuscript mentions temperature, relative humidity, and chemical characteristics of soil as environmental predictors. However, it is not clear how and where these environmental data were obtained. Were they measured in situ at each sampling location, retrieved from public databases, or modeled?

Results

Line 187 – indicate the total number of liver, spleen, lung, and feces samples

Line 211 - The statement that this is the first documented evidence of Histoplasma capsulatum infection in two opossum, six rodent and 14 bat species species is inaccurate. Which species? Please name the hosts even mnentioning Table 1.

Line 221-230 –

While the use of short sequences in phylogenetic analyses is often challenging due to limited phylogenetic signal, it is not inherently problematic when handled with appropriate methodological rigor. In this context, I recommend that the authors explicitly state the total number of Hc100 sequences used for tree construction and discuss potential limitations associated with sequence length.

To independently evaluate the robustness of the presented phylogeny, I reconstructed a maximum likelihood tree using IQ-TREE2 with the command -m TEST and support values estimated by both aBayes and 1,000 ultrafast bootstrap replicates. The Ecuadorian strains appear to cluster into two clades; however, the branching pattern between these groups may be artefactual or reflect insufficient resolution.

The presented tree herein discussed shows two clades. Notably, the upper clade (labeled in blue), which contains the majority of the sequences, clusters more closely with the reference strain H67 (associated with the LAm A2 lineage; see Kasuga et al., 2003). In contrast, the lower clade (yellow) is more closely related to strain H66, which has been associated with the LAm B lineage (Kasuga et al., 2003). Still there is a bias on the sequence length here. I strongly encourage the authors to re-express the phylogenetic relationships in light of these findings and consult prior literature that more comprehensively characterizes these lineages—particularly the work https://pubmed.ncbi.nlm.nih.gov/31372546/, which provides deeper insights into the genetic structure and population differentiation of colombian strains and is comparable to your dataset. See attached files

Another possibility is mention that strains are grouped into two different phylogenetic clusters without mentioning their genetic background.

I have no concerns about the species niche modeling.

Discussion

The discussion is generally well-written and informative; however, it would benefit from a deeper contextualization of the findings within a One Health framework. In particular, I recommend expanding the section to include more information on the burden and epidemiology of histoplasmosis in humans in Ecuador and neighboring regions. There is ample evidence that histoplasmosis is endemic in Ecuador, especially in immunocompromised individuals, including people living with HIV. Integrating this information would help situate the findings on opossum infection within a broader ecological and public health context, supporting the relevance of the study to One Health initiatives and pathogen surveillance strategies in the region.

Reviewer #2: While I think this could be a great contribution to the literature on Histoplasma in Ecuador, it needs significant improvement. In the introduction, there is a statement that birds and bats transmit Histoplasma via fecal route. While it is true that guano piles in caves and coops have been implicated sources of infection during outbreaks, there is still much debate on carriage. For the few studies that have directly collected from living animals, the rate of positivity is still low or zero. Dead animals are also found in guano piles thus Histoplasma could grow out of infected tissue, rather than dispersed via fecal route.

Reviewer #3: In regarding to the revision of the Manuscript MS PNTD-D-25-0030, entitled

"Histoplasma capsulatum in Wild Mammals from Ecuador", I would like to

emphasize its importance in the epidemiology of histoplasmosis, mainly

considering a wide sort of animals studied (106 species) and the role of wild

mammals as possible reservoirs and, particularly, the role of bats as the main

reservoir and disperser of the fungal pathogen H. capsulatum in nature, as have

been suggested by some researchers. Thus, any type of evidence for the

occurrence of H. capsulatum in wild mammals, either in Latin America or in other

regions or the world should be privileged. However, I would like to question the

efficacy of the Hcp100 molecular marker for the phylogenetic analysis. The Hcp100

molecular marker is successfully used in clinical diagnosis and for searching the H.

capsulatum presence in randomly infected animals, because this marker amplified

a specific DNA fragment of 210 bp, in the second Nested-PCR reaction, which is

unique for this fungal pathogen. Although its use for H. capsulatum phylogenetic

analyses is not the most recommendable, mainly considering that the Hcp100

fragment presents very low polymorphism, clades obtained were certainly

representative and discriminate the different groups of animals studied. Hence, I

suggest including appropriate interpretations in the Discussion section of the

Manuscript.

Overall, considering the above-comment, I think that the Manuscript must be

accepted.

PLOS authors have the option to publish the peer review history of their article (what does this mean? ). If published, this will include your full peer review and any attached files.

**Do you want your identity to be public for this peer review?** For information about this choice, including consent withdrawal, please see our Privacy Policy .

Reviewer #1: No

Reviewer #2: No

Reviewer #3: No

**Figure resubmission:**

**Reproducibility:**



---

## [Editor Report · Decision Letter 1]

30 Jul 2025

Dear Dr. Bastidas,

We are pleased to inform you that your manuscript 'Histoplasma capsulatum in Wild Mammals from Ecuador' has been provisionally accepted for publication in PLOS Neglected Tropical Diseases.

Best regards,

Angel Gonzalez, Ph.D.

Academic Editor

Marcio Rodrigues

Section Editor

Shaden Kamhawi

co-Editor-in-Chief

Paul Brindley

co-Editor-in-Chief

---

## [Editor Report · Acceptance letter]

Dear Sr. Bastidas-Caldes,

We are delighted to inform you that your manuscript, "Histoplasma capsulatum in Wild Mammals from Ecuador," has been formally accepted for publication in PLOS Neglected Tropical Diseases.

Best regards,

Shaden Kamhawi

co-Editor-in-Chief

Paul Brindley

co-Editor-in-Chief
